# Antibiotic-induced gut microbiome dysbiosis: risks and strategies for mitigation

Katharina Müller [ID] [1,3], Justine Gillard [ID] [1,3], Athanasios Typas [ID] [2], Michael Knopp [ID] [2✉] & Camille V Goemans [ID] [1✉]

## Abstract

**The discovery of antibiotics and their subsequent therapeutic use revolutionized our ability to treat once deadly infectious diseases, and antibiotics have become one of the most commonly prescribed drug classes. Unfortunately, these compounds not only target pathogenic strains, but also non-pathogenic bacteria that fulfill important functions for the human host. As such, antibiotic treatment can cause severe collateral damage, resulting in dysbiosis, for example, in the human gut microbiome. Given the immense importance of the gut microbiome for human health, antibiotic-induced dysbiosis can cause a variety of detrimental health outcomes. In addition, antibiotic (over-)use causes selection of antibiotic-resistant strains, and the human gut microbiome has become a major reservoir for resistance determinants that can transfer to pathogenic isolates and cause hard-to-treat infections. In this review, we describe various adverse effects that antibiotic use has on the human gut microbiome, how we can approach this problem experimentally, and discuss pathways to mitigate antibiotic-induced collateral damage.**

**Subject Category** Microbiology, Virology & Host Pathogen Interaction

## How antibiotics affect the gut microbiota composition

### The target spectrum of antibiotics on gut bacteria

Antibiotics are molecules that either kill or inhibit the growth of target bacteria. Given their immense importance for modern medicine, the antibiotic mode of action has been extensively studied, mostly on laboratory model species or common pathogens that are targeted by drug therapy. As the majority of gut microbes are phylogenetically distant from commonly studied organisms (Almeida et al, 2019), it is often difficult, if not impossible, to generalize the physiological response to antibiotic pressure from one species to another. As such, gut bacteria can have different susceptibility profiles to antibiotics, which are rarely in line with those of commonly used model bacteria. Intrinsically resistant bacteria, i.e., bacteria that are naturally resistant to a given antibiotic (for example, because of the absence of the antibiotic target), can tolerate much higher levels of antibiotics than susceptible counterparts. In addition, susceptible bacteria can develop antibiotic resistance by acquiring resistance-conferring mutations or bona fide resistance genes, causing largely different resistance profiles, even for closely related species or strains, based on the presence or absence of such resistance determinants. Due to these differences in antibiotic susceptibility, members of the gut microbiota community respond differently to antibiotic treatment, which can ultimately lead to dysbiosis (i.e., changes in bacterial composition and a decrease in bacterial diversity).

### Understanding the impact of antibiotics on the gut microbiota composition in vivo

The available literature describing the impact of antibiotics on the healthy human microbiota is limited. A handful of studies, mostly based on 16S rRNA sequencing, describe the effect of diverse antibiotics on the microbiota of healthy individuals (Korpela et al, 2020; Dethlefsen and Relman, 2011; Dethlefsen et al, 2008; De La Cochetière et al, 2005; Mangin et al, 2012; Chopyk et al, 2023; Choo et al, 2023). In all cases, the authors observed an acute decrease in taxonomic diversity and in some cases, incomplete recovery after treatment. For a more detailed analysis of the effects of specific antibiotic classes on gut bacterial taxa, we refer the reader to this recent review (Fishbein et al, 2023). In general, the magnitude and duration of antibiotic effects vary highly between individuals, but the adult microbiota usually displays resilience after antibiotic perturbation. Repetitive disturbances however can lead to more permanent composition shifts (Rashidi et al, 2021; Dethlefsen and Relman, 2011; Dethlefsen et al, 2008; De La Cochetière et al, 2005; Mangin et al, 2012). Using metagenomics, more recent studies compared the response of healthy adults to antibiotics and showed that treatment led to a decrease in species richness with most microbiomes returning to pre-treatment species richness after several months, but with an altered taxonomy, resistome and metabolic output (Xue et al, 2023; Anthony et al, 2022; Zaura et al, 2015; Bhattarai et al, 2024; d'Humières et al, 2024; Dhariwal et al, 2023; Benitez et al, 2025; Patangia et al, 2024; Hong et al, 2024).

---

[1]Global Health Institute, School of Life Sciences, Ecole Polytechnique Federale de Lausanne, Lausanne, Switzerland. [2]Molecular Systems Biology Unit, European Molecular Biology Laboratory, Heidelberg, Germany. [3]These authors contributed equally: Katharina Müller, Justine Gillard. ✉E-mail: knopp@embl.de; camille.goemans@epfl.ch

**Table 1.  Advantages and disadvantages of different experimental methods to study antibiotics impact on the human gut microbiota.**

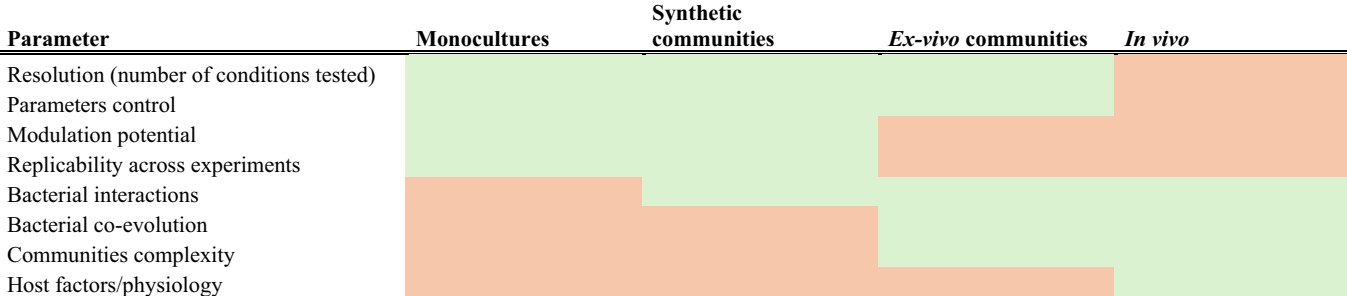

| Parameter | Monocultures | Synthetic communities | Ex-vivo communities | In vivo |
|---|---|---|---|---|
| Resolution (number of conditions tested) | | | | |
| Parameters control | | | | |
| Modulation potential | | | | |
| Replicability across experiments | | | | |
| Bacterial interactions | | | | |
| Bacterial co-evolution | | | | |
| Communities complexity | | | | |
| Host factors/physiology | | | | |

Green indicates high potential, while orange marks low potential.

Our understanding of the recovery post-antibiotic treatment is limited and can be impacted by a variety of complex factors, such as the initial diversity, diet, or environmental reservoirs (Ng et al, 2019). In addition, it has been shown to strongly vary between individuals (Raymond et al, 2016; Dethlefsen and Relman, 2011; Zaura et al, 2015; Pérez-Cobas et al, 2013). Species or metabolic functions influencing gut microbiota recovery after treatment are currently unknown. A study comparing microbiomes from 117 individuals identified 21 bacterial species exhibiting robust association with microbiome recovery post antibiotic treatment. These species were found to metabolize both host- and diet-derived energy sources, with their degradation products facilitating the repopulation of other microbes, a function further validated in mice (Chng et al, 2020).

Antibiotics are heavily prescribed, with the highest prescription rate for children below 2 years old, mostly for broad-spectrum antibiotics such as beta-lactams and macrolides (Vaz et al, 2014). As this stage is critical for the development of the intestinal microbiota, antibiotic treatment during that period leads to long-term microbial community changes and to an increased risk for diseases (Reyman et al, 2022; Cox et al, 2014). Opportunistic infections are the most common early complications of microbiome disruption (Gonzalez-Perez et al, 2016; Zhou et al, 2020; Shao et al, 2019; Deshmukh et al, 2014; Sekirov et al, 2008; Stevens et al, 2011; Yip et al, 2023). A healthy microbiota protects its host from pathogen invasion either by direct competition (production of antimicrobial effectors or competition for space and nutrients) or indirectly by eliciting an immune response (Abt and Pamer, 2014; Brown and Clarke, 2017; Iizumi et al, 2016; Buffie et al, 2012; Gonzalez-Perez et al, 2016; Zhou et al, 2020; Shao et al, 2019; Sekirov et al, 2008; Woelfel et al, 2024). A number of model studies have shown that antibiotic exposure, by altering the composition of the microbiota, increases the severity of enteric infections in adult animals including humans (Isaac et al, 2017; Ubeda et al, 2010; Buffie and Pamer, 2013), an effect that is conserved when the altered microbiota is transferred to antibiotic-naive mice (Roubaud-Baudron et al, 2019; Sorbara et al, 2019; Becattini et al, 2017; Wlodarska et al, 2011; Barthel et al, 2003; Stecher et al, 2007). Antibiotic treatment of mice was, for example, shown to induce a spike in microbiota-liberated sialic acids. *Salmonella enterica* serovar Typhimurium and *Clostridioides difficile*, two antibiotic-associated opportunistic pathogens, depend on these carbohydrates and metabolize them during their expansion within the gut (Ng

et al, 2013a). In addition to opportunistic infections, children cohort and mice studies (Korpela et al, 2016; Aversa et al, 2021; Nguyen et al, 2020) show that early-in-life antibiotic treatment is associated with long-lasting developmental, immunological, and metabolic changes. It has been associated with an increased risk of developing childhood onset asthma (Russell et al, 2012; Borbet et al, 2019; Stokholm et al, 2018; Metsälä et al, 2015), allergies (Metsälä et al, 2013), obesity, type II diabetes and related metabolic diseases (Mahana et al, 2016; Cox et al, 2014; Nobel et al, 2015; Cho et al, 2012; Cox and Blaser, 2015; Petschow et al, 2013; Schulfer et al, 2019), type I diabetes (Zhang et al, 2018; Stewart et al, 2018; Horton and Blaser, 2020; Mikkelsen et al, 2015; Livanos et al, 2016), kidney stones (Tasian et al, 2018), immunological and auto-immune diseases (Zhang et al, 2021; Ruiz et al, 2017; Abt et al, 2012; Macpherson and Uhr, 2004; Arrieta et al, 2016; Kemppainen et al, 2017), inflammatory bowel disease, colitis or celiac disease (Hviid et al, 2011; Kronman et al, 2012; Schulfer et al, 2017; Ruiz et al, 2017; Ozkul et al, 2020; Ng et al, 2013b), with macrolide and lincosamide antibiotics exhibiting the strongest effects (Nobel et al, 2015; Anthony et al, 2022; Zaura et al, 2015; Korpela et al, 2016).

Studying the impact of antibiotics on the microbiota in vivo is valuable, as it reflects the microbiota's physiological niche; however, it also presents several limitations: (i) there is a lack of standardization of the methods, which leads to inconsistencies across studies; (ii) the antibiotics impact is generally considered at the genus or species levels, while there is an increasing awareness that strain-to-strain variation is critical; (iii) human-based studies are typically descriptive and only allow for association-based conclusions and (iv) findings from in vivo studies in mice are difficult to translate to humans because of experimental design, differences in host biology and in natural microbiome composition (Table 1).

## Increasing our prediction power, from in vitro monocultures to communities

The use of in vitro approaches has dramatically increased the number of drugs with known impact on gut bacteria (Ecale et al, 2021; Maier et al, 2018, 2021). By studying the effect of 144 antibiotics on 40 gut microbes in monoculture, we previously built a high-resolution map of the effect of antibiotics on those microbes. Sensitivity profiles were highly species- and sometimes strain-specific, and phylogeny did not infer sensitivity (Maier et al, 2021).

In addition, our in vitro observations provided a hypothesis to explain the strong effect of macrolides observed in human cohorts or in mice (Nobel et al, 2015; Anthony et al, 2022; Zaura et al, 2015; Korpela et al, 2016). Macrolides and tetracyclines, both textbook bacteriostatic antibiotics, were shown to inhibit the growth of all commensals tested but also killed some of those species, exhibiting a species-dependent bactericidal activity. Bacterial isolates that were killed by these antibiotics were more readily eliminated from synthetic communities than those for which only growth was prevented, thereby inducing dysbiosis (Maier et al, 2021) (Table 1).

Can data acquired from bacteria grown in monoculture predict the impact of antibiotics on highly complex bacterial communities? To start addressing this question, different groups have combined in vitro and ex vivo approaches to study synthetic (bottom-up assembled) or natural (stool-derived) communities. The concept that one species could alter the sensitivity of another species had already been described in the 70 s, but was never tested systematically (Onderdonk et al, 1979). An effort in that direction was made by comparing the response from isolated gut microbial species versus their response in a synthetic community, upon treatment with commonly used drugs (Garcia-Santamarina et al, 2024). This study shows that in 75% of the cases, the bacterial presence in communities after antibiotic treatment can be inferred from its sensitivity in monoculture. In 25% of the cases, however, new interactions emerged in communities, such as cross-protection of drug-sensitive species and cross-sensitization of drug-resistant species (Garcia-Santamarina et al, 2024). Cross-protection mechanisms are more frequent, which implies that bacterial communities tend to be more resilient than species alone (Garcia-Santamarina et al, 2024). A recent example of cross-protection shows that the presence of *Acetobacter* species increases the antibiotic tolerance of *Lactobacillus plantarum* by consuming the lactate that the latter produces. This decreases the acidification of the medium, which correlates with antibiotic tolerance (Aranda-Díaz et al, 2020). Cases of cross-sensitization have also been demonstrated, for example, in communities with obligate cross-feeding, i.e., when one strain relies on a metabolic product produced by another strain to grow. In such cases, the minimal inhibitory concentration (MIC) of both strains from the community drops to that of the producing strain. This was tested in an obligate cross-feeding system, engineered with *Escherichia coli, Salmonella enterica*, and *Methylobacterium extorquens.* The authors observed that, as expected, resistant species were inhibited at significantly lower antibiotic concentrations in the obligate cross-feeding community than in monoculture (Adamowicz et al, 2018). More generally, any type of competition (incl. resource competition) or dependency between members of a community will influence the composition of the community after treatment (Ho et al, 2024; Newton et al, 2023).

To homogenize results across studies, standard microbiota communities have been established over the last few years. For example, the OMM-12 is a murine gut community covering the major bacterial phyla that has been used to understand community dynamics, antibiotic impact, and protection from pathogens (Brugiroux et al, 2016; Eberl et al, 2020). As synthetic communities typically lack the complexity of natural microbiota, more complex synthetic communities were established, such as the hCom1 and hCom2 (with >100 bacterial members) (Cheng et al, 2022). In parallel, controlled synthetic communities with species and strains isolated from a single complex microbiota (Aranda-Díaz et al,

2022) were assembled to mix only bacteria that previously co-evolved, as this could impact the resilience of those communities to antibiotic treatment. These stool-derived in vitro communities (SICs) can be used to model response to antibiotic treatment, pathogen invasion, or diet impact in vitro and in vivo (Aranda-Díaz et al, 2025, 2022). For example, a SIC derived from ciprofloxacin-treated mice showed increased *S. enterica* colonization in vitro. The compositional changes observed in SICs or in humanized mice were similar (Aranda-Díaz et al, 2022). In general, assembling communities to probe antibiotic action offers great potential but scientists always face a trade-off between complexity, diversity, coevolution, or dynamics and proper control of community composition (reviewed in (Shetty et al, 2019; Giri et al, 2025)) (Table 1).

## Protecting the gut microbiota from antibiotic treatment

One strategy to reduce the collateral damage of antibiotics is to employ molecules with a narrow target spectrum. Fidaxomicin, for example, is now the first-line treatment for *C. difficile* infection (CDI), as it was shown to only poorly inhibit the normal gut microbiota and to reduce the risk of recurrence compared to vancomycin (Cao et al, 2022; Zhanel et al, 2015). Ridinilazole (Mason et al, 2023; Collins and Riley, 2022; Okhuysen et al, 2024) and Ibezapolstat (Bassères et al, 2024; McPherson et al, 2022) were also recently developed to target CDI, and their clinical potential is currently being evaluated. Some non-antibiotic drugs with antimicrobial activity (Maier et al, 2018) could also be repurposed to target specific pathogens. Ebselen, for example, reduces colitis and recurrence risk of CDI without altering microbiome diversity through its anti-inflammatory and anti-toxin functions (reviewed in (Maślanka and Mucha, 2023; Collins and Riley, 2022)). Another way to reduce the antibiotic target spectrum is by the utilization of combination therapy, including multiple antibiotics, as the target spectrum of drug combinations has previously been shown to often be species-specific (Brochado et al, 2018; Cacace et al, 2023). A screen of over 1200 FDA-approved drugs led to the discovery of molecules that, taken together with antibiotics, protect gut microbes but not pathogens from antibiotics (Maier et al, 2021) (Fig. 1). As alternative to antibiotics, bacteriophages (i.e., viruses that infect and kill bacteria) or lantibiotics (i.e., ribosomally synthesized and post-translationally modified antimicrobial peptides) are promising antibacterial candidates, as they typically have a very narrow spectrum, thereby lowering the risk for dysbiosis. However, their true impact on the gut microbiota has only been poorly explored to date (Weirauch et al, 2026; Cole et al, 2025).

In most cases, antibiotics are taken to cure an infection outside of the gut. Therefore, an alternative strategy is to reduce the gut antibiotic concentration, which can be achieved by improved targeted delivery of the drug to the site of infection, or by removal/inactivation of antibiotics reaching the human gut. Targeted drug delivery aims at releasing the drug in a specific body compartment, thereby sparing the gut microbiome community. This can be achieved, for example, by using specific coatings, nanocarriers, or working with pro-drugs that are only activated by enzymes present in the target tissue. Another strategy is to remove or inactivate antibiotics in the colon. This is currently achieved with drug-

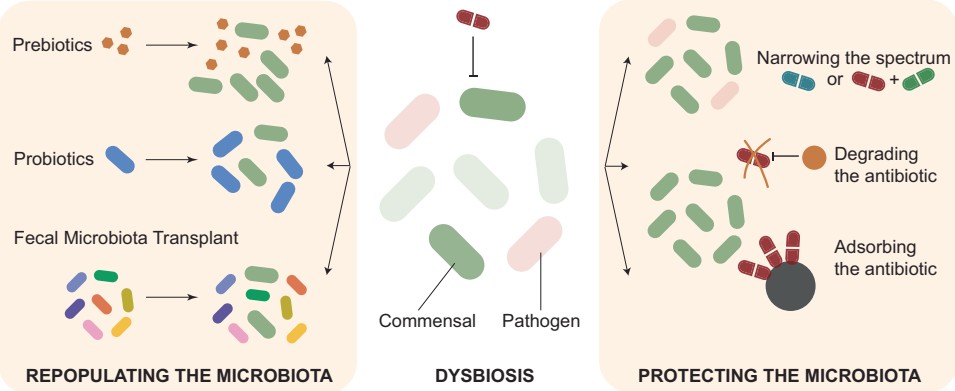

**Figure 1. How to protect the microbiota from antibiotic-induced dysbiosis?**

The current strategies to reduce the collateral damage of antibiotics on the human gut microbiota are to (i) narrow the drug activity spectrum by designing novel drugs or combining existing ones, (ii) to degrade the antibiotic in the colon, or (iii) to use adsorbing agents that will decrease the concentration of antibiotics. (ii) and (iii) are only useful if the infection occurs at a different body site. After dysbiosis, prebiotics, probiotics, or a fecal microbiota transplant can help repopulate the microbiota.

degrading enzymes or adsorbing agents. Ribaxamase is an orally administered β-lactamase that degrades penicillin and cephalosporin antibiotics. It alleviates ceftriaxone-mediated dysbiosis and attenuates the propagation of antibiotic-resistance genes (ARGs) in pigs (Connelly et al, 2017). In addition, it reduces the incidence of CDI in humans after ceftriaxone treatment (Kokai-Kun et al, 2019). A new formulation that allows its release in the lower intestine, SYN-007, was shown to protect gut microbiota during amoxicillin treatment in dogs: dysbiosis was alleviated, and the number of resistance genes was decreased, while the serum concentration of amoxicillin was unchanged (Connelly et al, 2019b). SYN-006, a carbapenemase isolated from *B. cereus* (Connelly et al, 2019c), was shown to prevent ertapenem-mediated dysbiosis in pigs and decreased the emergence of antibiotic resistance, while the serum levels of the antibiotic remained unchanged (Connelly et al, 2019a). Probiotics can also be engineered to degrade unwanted antibiotics in the gut. For example, *Lactococcus lactis* was engineered to secrete a β-lactamase, encoded by two genetically unrelated genes, to lower the probability of horizontal gene transfer. This probiotic was shown to reduce ampicillin-mediated dysbiosis in mice without affecting the systemic concentration, preventing the enrichment of antimicrobial resistance (AMR) genes in the gut microbiome and the loss of colonization resistance against *C. difficile* (Cubillos-Ruiz et al, 2022). Outer membrane vesicles produced from *B. thetaiotaomicron* and other *Bacteroides* species carry surface-associated β-lactamases able to degrade cefotaxime. Those OMVs were shown to protect *Salmonella* Typhimurium and *B. breve* from cefotaxime treatment (Stentz et al, 2015). While promising, these approaches can also be dangerous, as antibiotic-resistance genes are artificially introduced in the microbiota. If this is not properly controlled, this could lead to the spread of antibiotic resistance through HGT or to increased dysbiosis after treatment, where the probiotic carrying the resistance element would disproportionally survive. Adsorbing agents, like DAV132, deliver activated charcoal to the colon. DAV132 selectively adsorbs drugs in the proximal colon without interfering with drug absorption in the small intestine (Vehreschild et al, 2022; De Gunzburg et al, 2015; Messaoudene et al, 2024). The co-administration of DAV132 with

moxifloxacin decreased the colon antibiotic concentration by 99%, while serum levels were not affected. The composition and diversity of the gut microbiota of DAV132-treated patients remained mostly unchanged (De Gunzburg et al, 2018). The non-specificity of activated charcoal can be discussed, as it could also adsorb essential biological molecules. More specific adsorbents have now been developed. For example, a polyethyleneglycol-based microparticle adsorbent was engineered to adsorb vancomycin and was shown to protect *Staphylococcus lentus* from vancomycin in vitro and in mice (Yuzuriha et al, 2020), and an anion exchange resin was used to adsorb the beta-lactam cefoperazone in order to protect the gut microbiota from cefoperazone-induced dysbiosis in mice (Li et al, 2021) (Fig. 1).

Finally, several strategies, using prebiotics, probiotics, and fecal microbiota transplant, aim at facilitating the recovery of the gut microbiota during/after treatment. Prebiotics are non-digestible food components (mostly fibers) that selectively stimulate the growth or activity of beneficial microorganisms in the gut, thereby helping maintain a healthy gut microbiome. Probiotics are live microorganisms that, when consumed in adequate amounts, can confer health benefits to the host, while fecal microbiota transplant (FMT) is a medical procedure in which processed stool from a healthy donor is transferred into the intestine of a recipient to restore a balanced gut microbiome. As those have been extensively reviewed in (Ghani et al, 2022; Szajewska et al, 2025; Suez et al, 2019), we decided not to discuss them here (Fig. 1).

## The gut microbiota as a reservoir for antimicrobial resistance elements

Apart from the previously described intrinsic resistance, which is the inherent reduced antibiotic susceptibility of certain bacterial species independent of genetic change, the human gut microbiome harbors a diverse reservoir of antibiotic-resistant bacteria, where the phenotype is mediated by the presence of distinct resistance determinants. These resistant strains can be selected de novo from a susceptible wild-type strain by spontaneous chromosomal

mutations that decrease the antibiotic susceptibility. In addition, resistance can be achieved by the acquisition of bona fide resistance genes. Bacteria carrying resistance genes enter the gastrointestinal tract from exogenous sources, for example, by consumption of food contaminated with resistant bacteria or *via* human-to-human transmission, and subsequently colonize the human gastrointestinal tract (Mughini-Gras et al, 2019). These genes are often mobilizable, and antibiotic pressure can select for their horizontal dissemination in the gut microbiome. Resistance genes can also come from endogenous microbiota members and be transferred under certain conditions (e.g., antibiotic exposure). Here, we will describe the basic principles of antibiotic-resistance evolution and dissemination in the human gastrointestinal tract, the prevalence of resistant pathogens and detection methods, as well as the impact of antibiotic pressure on the dissemination of antibiotic-resistance elements to pathogenic bacteria. Finally, we will provide a perspective on how the gut microbiome could potentially be manipulated to hamper the spread of antibiotic-resistant bacteria.

## Chromosomal mutations

Upon antibiotic exposure, and for almost all antibiotics to date, spontaneous mutants with reduced susceptibility can emerge. The underlying mechanisms vary depending on the bacterial species and/or antibiotic used, but common resistance mutations alter the susceptibility of the drug target or reduce the concentration of the antibiotic in proximity to the target. Fluoroquinolone resistance mediated by target alteration mutations in *gyrA* or *parC* is pervasive in a wide range of bacterial species colonizing the human gastrointestinal tract (Yaffe et al, 2025), including non-pathogenic commensals such as *Lactobacillus* (Li et al, 2015a) as well as major bacterial pathogens such as *E. coli*, *Salmonella*, and *Klebsiella pneumoniae* (Ruiz, 2003). In addition, major gastrointestinal pathogens, such as *E. coli* or *Salmonella* Typhimurium, frequently carry mutations in negative regulators (such as *marR* and *acrR*), causing upregulation of the antibiotic efflux, or mutations in major outer membrane porins, causing a reduction in antibiotic influx (Nikaido, 2003; Li et al, 2015b; Weston et al, 2018).

## Resistance genes

Besides spontaneous mutations that activate intrinsic resistance determinants, bacteria can become resistant by the acquisition of bona fide resistance genes. The increase in resistance *via* these genes is mainly achieved by enzymatic degradation/modification of antibiotics, protection of the antibiotic target active, or efflux of antibiotics. Antibiotic degradation and target protection typically cause resistance to a narrow range of antibiotics, while efflux pumps can vary in their substrate specificity. Especially efflux pumps with a wider substrate specificity are of increased clinical importance, as their acquisition by a bacterial pathogen will cause multidrug resistance (Blair et al, 2015). The "resistome" composition of diverse environments such as humans, animals, waste-, fresh- or sea-water, soil, or air has been described extensively (Larsson and Flach, 2022; Allen et al, 2010; Zhuang et al, 2021). ARGs have been isolated from pristine and ancestral environments, demonstrating that resistance predates clinical antibiotic use (Van Goethem et al, 2018; D'Costa et al, 2011). While ARGs in these ancient samples were generally not mobile (Van Goethem et al,

2018; D'Costa et al, 2011), Cazares et al highlighted a critical shift following the introduction of antibiotics with the acquisition and evolution of ARGs on plasmids, which is a principal mechanism for their dissemination (Cazares et al, 2025). Consequently, anthropogenic activities, such as clinical use of antibiotics, have a strong influence on the abundance of these genes (Zhang et al, 2022).

Many resistance genes are chromosomally encoded and inherent to distinct bacterial species (Diebold et al, 2023), while others are encoded on mobile genetic elements (MGEs), which allows the horizontal intra- and inter-species transfer. MGEs undergo multiple horizontal gene transfer mechanisms, including transduction via bacteriophage infection and conjugation, i.e., the contact-dependent exchange via a pilus of elements such as plasmids, integrative and conjugative elements, transposons, and integrons. The "mobilome" plays a central role in the propagation of antibiotic resistance, as approximately one-third of the human gut resistome is associated with at least one class of MGE (Lee et al, 2023).To illustrate, conjugative plasmids often harbor a large repertoire of resistance genes that can be co-localized with transposable elements, allowing rapid dissemination and structural rearrangements in response to antibiotic pressure (Rajer et al, 2022). Densely populated environments that contain opportunistic pathogens and frequently encounter antibiotics, such as the human gastrointestinal tract, serve as problematic melting pots for the transfer of antibiotic-resistant genes to bacteria that can subsequently cause hard-to-treat infections (Gumpert et al, 2017; Kent et al, 2020).

## Detection of antibiotic resistance

There is no "gold standard" for the detection of antibiotic resistance, but several approaches have been employed to identify and/or characterize ARGs in the human gut, each with its own limitations and advantages. Culture-based approaches are among the most frequently employed in diagnostics to identify antibiotic-resistant bacterial isolates. While this method is easy, cheap, and does not require any advanced experimental platforms, it is restricted to the bacterial strains for which the chosen culture method enables growth. Considering that most bacterial species are difficult to cultivate and that different culture media support the growth of only selective bacteria, these culture-based approaches are not well-suited for an unbiased detection of resistance elements in a complex microbiome sample, but rather for the identification of resistant isolates of a specific bacterial species. Furthermore, culture-based approaches show antibiotic resistance at a phenotypic level (which is the most relevant information in clinical microbiology when deciding on treatment regimens), but it does not establish which resistance gene(s) are causing the increase in antibiotic resistance.

A method that circumvents several of these shortcomings is the detection of ARGs from complex microbiome samples using functional metagenomics, where the total DNA of the microbial community is extracted, sheared, and cloned into a synthetic expression vector. This vector library is then screened in a strain of interest for increased levels of resistance to antibiotics. This approach circumvents two limitations from culture-based susceptibility testing: (i) the library can be screened in a single, well-defined strain of interest that shows robust growth in the chosen assay medium and has a defined susceptibility to the antibiotics of interest, and (ii) an increase in resistance is mediated by

overexpression of a single genetic locus. However, this approach is not only tedious, but it also requires any potential resistance element to be functional in the strain used for overexpression, and it must be active, yet not toxic, under the artificially set expression levels. While most resistance elements are active in diverse species, some do rely on a specific genetic backbone and cannot be easily detected using functional genomics.

An alternative approach in detecting ARGs that does not rely on culturing and/or functionality in a focal strain is based on metagenomics. Here, a given set of metagenomes can be easily screened for the presence or absence of previously described resistance genes. This method allows the identification and quantification of resistance elements of interest in silico without strain bias or the need to culture. Naturally, this approach is limited to resistance elements that have already been described before, and it also does not comprehensively determine if or how much the identified resistance element increases antibiotic resistance in the bacterial isolate carrying it. Furthermore, while bona fide resistance genes can generally be identified with high confidence using comparative genomics, it is often difficult to differentiate between natural sequence variation in certain genes associated with antibiotic resistance to mutations that have been selected due to antibiotic pressure. Another shortcoming is that resistance genes localized within mobile genetic elements can usually not be linked to the host organism. Certain techniques, such as Hi-C cross-linking, have been developed to establish an association between host bacteria and resistance determinants (Kent et al, 2020; Stalder et al, 2019), however, the sensitivity is limited by sequencing depth and thus low-abundance resistance determinants in highly complex samples are not easily accessible.

## Prevalence of antibiotic resistance in the human gut microbiome

The prevalence of ARGs in the human gut microbiome has been extensively studied. Even though the above-described limitations of detection prevent a precise estimation of resistance gene abundance in complex microbiomes, the human gut serves as a major reservoir of resistance determinants to all major classes of antibiotics (Crits-Christoph et al, 2022; Lee et al, 2023; Ho et al, 2020; Lamberte and Van Schaik, 2022; Anthony et al, 2021; Wallace et al, 2020). A metagenome-wide analysis of 162 individuals found that ARGs make up ~1:4000 of the total gut microbial genes (Hu et al, 2013). Similarly, studies reported a widespread distribution of AMR genes in fecal samples of healthy individuals, with tetracycline resistance being the most abundant resistance gene, followed by fluoroquinolone (Feng et al, 2018; Li et al, 2023). While the high prevalence of AMR genes is certainly alarming (Shoer et al, 2024), the medical burden is imposed by the presence of these resistance genes in pathogenic bacteria. One prominent representative of this group is *E. coli*, which is present in the gastrointestinal tract of all humans, yet represents a major human pathogen causing the majority of urinary tract infections (Flores-Mireles et al, 2015). Particularly hard to treat are infections by *E. coli* strains producing extended-spectrum beta-lactamases (ESBL), which render the bacteria resistant to a majority of clinically relevant beta-lactam antibiotics (Paterson and Bonomo, 2005). While colonization by ESBL-producing *E. coli* is mostly asymptomatic, it can serve as a reservoir and has been linked to subsequent infection (Ruppé et al, 2013;

Reddy et al, 2007), especially for recurrent urinary tract infections (Worby et al, 2022). Similar to the pervasive spread of resistance genes in the human gut microbiome, a systematic study by Bezabih et al screening 62 research articles covering ~30,000 healthy people showed a 16.5% intestinal carriage of ESBL-producing *E. coli*, with an increasing rate over the last 15 years (Bezabih et al, 2021). In addition to known antibiotic-resistance determinants, the human gut microbiome is expected to harbor yet uncharacterized resistance determinants. One estimation based on three-dimensional structure homology comparative modeling predicts over 6000 antibiotic-resistance determinants that were only distantly related to known resistance elements (Ruppé et al, 2018).

The abundance of resistance determinants in the human gastrointestinal microbiome is affected by several biotic and abiotic factors. As expected, it is strongly altered upon, or shortly after, antibiotic treatment. Several studies have compared antibiotic-resistance load in the gut microbiome before and after antibiotic treatment and showed that antibiotics not only decrease species richness but also enrich resistance gene abundance and in some cases carriage of multidrug-resistant opportunistic pathogens, even after recovery of microbiota composition (Gasparrini et al, 2019; Anthony et al, 2022; Palleja et al, 2018; Yaffe et al, 2025; Fredriksen et al, 2023). By contrast, in the Schluter et al study, antibiotics reduced the diversity of both the gut bacterial community and the resistance gene pool it carries (Schluter et al, 2023). Another major determinant of ARG carriage is the geographical location of the human host, since antibiotic usage, and therewith presence of trace amounts of antibiotics in food, terrestrial or aquatic environments, is vastly different between different countries (Boolchandani et al, 2022; Arcilla et al, 2017; Bengtsson-Palme et al, 2015). While country-specific differences in the human gut resistome have been linked, for example, to different practices in agricultural and clinical antibiotic use (Forslund et al, 2013), differences have also been linked to a variety of factors such as diet (Pärnänen et al, 2025; Oliver et al, 2022; Stege et al, 2022), non-antibiotic drug intake (Lee et al, 2024; Ding et al, 2025), genetic factors (Salehi et al, 2025; Li et al, 2020; Goodrich et al, 2014) or pet ownership (Anthony et al, 2021). Even though a direct comparison of different studies dissecting factors driving the human gut resistome is often limited by experimental heterogeneity (Ho et al, 2020), it is established that the human gut not only harbors a plethora of ARGs, but that it also serves as a reservoir for infections caused by antibiotic-resistant pathogens.

## Deleterious effects caused by antibiotic resistance

While antibiotic pressure imposes a strong selection for antibiotic-resistant mutants, the mechanisms underlying the resistance are often associated with adverse effects on the bacterial physiology (Andersson and Hughes, 2010; Andersson, 2006). For example, mutations in the earlier described outer membrane proteins OmpC/OmpF in *E. coli* (OmpK35/OmpK36 in *Klebsiella pneumoniae*) not only cause a reduction in antibiotic import, but simultaneously can reduce the cellular nutritional competence (Phan and Ferenci, 2017), which can result in a reduction in the exponential growth rate (Knopp and Andersson, 2015). Even resistance mutations that do not show a significant effect on bacterial growth rate, such as in *gyrA* (Knopp and Andersson, 2018), can have deleterious fitness effects, for example, in the form

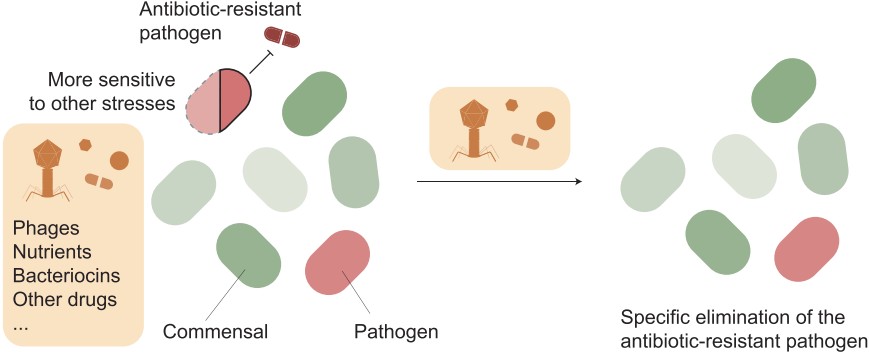

**Figure 2. How to counter-select resistance in the gut?**

Antibiotic resistance, while providing a fitness advantage to its host during antibiotic stress, can decrease its fitness under other conditions by making it more sensitive to other stresses. These stresses include phage predation, bacteriocin-mediated competition, nutrient intake, or drug treatment. This means that, upon exposure to these stresses, antibiotic-resistant bacteria can be specifically eliminated when in the right community or condition.

of reduced virulence (Tsai et al, 2011; Sánchez-Céspedes et al, 2015). Similarly, the expression of resistance genes can exert a physiological disruption to the cell, such as expression of ß-lactamases that alter the fraction of cross-linked muropeptides (Fernández et al, 2012) or efflux pumps that can cause cytoplasmic acidification (Olivares Pacheco et al, 2017). The "fitness effect" associated with antibiotic-resistance determinants is naturally dependent on various parameters, including the host genetic background (Knopp and Andersson, 2018) or environmental factors (Hubbard et al, 2019). Similar to these described "conditional fitness effects", the acquisition of resistance genes can sensitize or desensitize the bacterial cell to other bioactive agents, such as antibiotics and non-antibiotic drugs (Wang et al, 2023; Maier et al, 2018; Imamovic and Sommer, 2013), antimicrobial peptides (Lázár et al, 2018) or phages (Burmeister et al, 2020), leading to collateral sensitivity or cross-resistance. Antibiotic-resistance-induced hypersusceptibility to other conditions represents a promising angle for the design of treatment regimens and has received increasing attention in the last decade, where novel antibiotics are scarce, and the repurposing of existing antibiotics to circumvent resistance becomes more and more attractive (Tyers and Wright, 2019; Brochado et al, 2018; Aulin et al, 2021) (Fig. 2).

### Perspective—can we exploit gut microbiome-specific fitness effects to counter-select antibiotic-resistant mutants?

As discussed above, the gut serves as an extensive reservoir of diverse antibiotic-resistance determinants harbored by both pathogenic and commensal bacteria. The phenotypic expression of most of these resistance determinants has been thoroughly characterized in vitro using monocultures. This is in stark contrast however to their natural environments, where resistant strains not only have to compete with a plethora of community members for nutrients, but also need to evade phages and toxic compounds produced by other bacteria. If a resistant mutant is more susceptible to certain toxic compounds it might encounter in complex communities (such as bacteriocins) or less efficient in the uptake of nutrients, the microbiome could potentially counterselect these resistant mutants, effectively reducing resistance gene burden and

therewith reducing the likelihood of an infection caused by a multidrug-resistant pathogen. In a pioneering study by Cardoso et al, it was shown that in an in vivo mouse model, the fitness of E. coli carrying chromosomal resistance mutations varied between different mice depending on dysbiosis of the residual host microbiota (Leónidas Cardoso et al, 2020). While this study is limited to a specific pair of resistance mutations and the molecular mechanisms affecting mutant fitness remain elusive, the study highlights the importance of community effects when investigating the fitness of resistance determinants. In our ongoing work, we have systematically screened the influence of the community composition on the fitness of antibiotic-resistant gut commensals and identified several cases where the fitness of the resistant strain was community-dependent. One such example caused the evolution of a high-fitness antibiotic-resistant K. pneumoniae mutant in a microbiome-dependent fashion. This evolution was driven by carbohydrate competition and strongly dependent on higher-order interactions within the microbiome (Knopp et al, 2026). Notably, the selected mutation was found to be widespread amongst clinical isolates of K. pneumoniae, showing that such in vitro screening approaches can capture selective pressures relevant in nature (Knopp et al, 2026). The development of screening platforms to probe these interactions at scale could be used to identify microbiome-specific drivers and inhibitors of antibiotic-resistance evolution and provide a novel avenue for intervention (Fig. 2).

## Peer review information

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

## Acknowledgements

This work was supported by the Swiss National Science Foundation Project funding grant (10002165) (KM), the Swiss National Science Foundation Starting Grant (TMSGI3_226334/1) (JG), and the postdoctoral research grant (2019-00666) of the Swedish Research Council (MK).

## Author contributions

**Katharina Müller**: Writing—original draft. **Justine Gillard**: Writing—original draft. **Athanasios Typas**: Supervision; Validation. **Michael Knopp**: Supervision; Funding acquisition; Writing—original draft. **Camille V Goemans**: Supervision; Funding acquisition; Writing—original draft.

## Disclosure and competing interests statement

The authors declare no competing interests.

