## [Peer Review File · The EMBO Journal]

Antibiotic-induced gut microbiome dysbiosis: Risks and strategies for mitigation

Katharina Mueller, Justine Gillard, Athanasios Typas, Michael Knopp, and Camille Goemans

Corresponding authors: Camille Goemans (camille.goemans@epfl.ch) , Michael Knopp (knopp@embl.de)

Review Timeline:

Submission Date:	6th Nov 25
Editorial Decision:	9th Feb 26
Revision Received:	31st Mar 26
Accepted:	22nd Apr 26

Editor: Ieva Gailite

Transaction Report:

Dear Camille,

Thank you for submitting your review article to The EMBO Journal. I sincerely apologise for the unusually prolonged assessment process for your manuscript due to the holiday period and the resulting delays in reviewer report submission. I have now received comments from two reviewers, which are attached below.

As you will see, both reviewers appreciate the topic and the timeliness of the review. In addition, they provide a couple of minor but constructive points for the improvement of the article. Therefore, I would like to invite you to submit a revised version of the manuscript with these suggestions incorporated. Regarding the comment of reviewer #2 about line 114, the text as written currently appears correct to me.

From the editorial side, I have also included in the attachment further details on figure preparation for the final version. I believe that the figures look rather clear and we might not need to involve an external graphics editor, but they might nevertheless be helpful.

There are also a few formatting requests that we would like to ask you to incorporate in the final version:

1. Please remove figures and the table from the manuscript text file, while leaving the figure legends in.
2. We are missing the ORCID iD for the co-corresponding author Michael Knopp. In order to link the ORCID iD to the account in our manuscript tracking system, the author in question has to do the following:
 - Click the 'Modify Profile' link at the bottom of your homepage in our system.
 - On the next page you will see a box halfway down the page titled ORCID*. Below this box is red text reading 'To Register/Link to ORCID, click here'. Please follow that link: you will be taken to ORCID where you can log in to your account (or create an account if you don't have one)
 - You will then be asked to authorise Wiley to access your ORCID information. Once you have approved the linking, you will be brought back to our manuscript system.Unfortunately, we cannot do this linking on the author's behalf for security reasons.
3. The funding information should be correct and identical both in the manuscript and our online system. Please add an "Acknowledgements" section to the manuscript text file listing the relevant funding sources, before "References".
4. Please add the "Disclosure and competing interest statement" section before "Acknowledgements".
5. Please update references according to The EMBO Journal style. The articles should be listed alphabetically. Where there are more than 10 authors on a paper, the first 10 should be listed, followed by 'et al.'

Please let me know if you have any questions about these points. Thank you for preparing such a timely review, and I look forward to receiving the revised article!

With best wishes,

Ieva

Read our guidance for manuscript revisions and related editorial policies: <https://link.springer.com/journal/44318/submission-guidelines#cms-Revised-submissions>

We realize that it is difficult to revise to a specific deadline. In the interest of protecting the conceptual advance provided by the work, we recommend a revision within 3 months (10th May 2026). Please discuss the revision progress ahead of this time with the editor if you require more time to complete the revisions.

Referee #1:

The review by Müller et al., focuses on implications of antibiotics on the gut microbiome and underlying mechanisms. Overall, I appreciate the breadth and depth of manuscript, but would suggest to expand the paragraphs on global differences in antibiotic usage and associations with dysbiosis to better align with the title of the paper. Are there indications that increased antibiotic usage is associated with depletion of key taxa in the population? Along the same line it may be prudent to discuss the presence of antibiotic resistance genes in more ancestral populations that may not yet have been exposed to antibiotics. The paper may also gain broader interest if there is a discussion/outlook on what the options for us a society can do since it is unlikely that antibiotic resistance genes will disappear even if the antibiotic is withdrawn. Beside reducing antibiotic usage there may be alternative approaches such as lantabiotics or phages.

Referee #2:

In the manuscript "Antibiotic-induced gut microbiome dysbiosis: Risks and strategies for mitigation," Muller and colleagues wrote a review on the recent literature on how antibiotics affect the microbiota and include a critical review on novel recent approaches focusing on understanding the mechanisms behind the effects of antibiotics on the microbiota community and the emergence of antibiotic resistance. They review the work showing how studies comparing laboratory cultures, monocultures, and communities can help identify novel mechanisms and predict the impact of antibiotics on gut communities in animal studies, and discuss the advantages and limitations of the different approaches to studying the impact of antibiotics on the gut microbiota. They also review the literature on how the gut microbiota can serve as a reservoir for antimicrobial resistance and discuss potential strategies to leverage the microbiota to counter-select for the emergence of antimicrobial resistance.

The review is well-written and timely.

I have only a couple of minor comments/suggestions that can help improve the manuscript:

- Rephrase the sentence on lines 34-35. Here, you should not be saying that bacteria can have different susceptibility profiles. You should change this sentence to make it explicit that gut bacteria can have susceptible profiles that are difficult to predict, given the profiles of typical model bacterial laboratory isolates.

- When referring to the role of the microbiota in direct competition to pathogen colonization (colonization resistance), lines 70-72, references to colonization resistance are missing. You can refer to the recent review by Stecher, or add some of the original work reference within that reference.

Intestinal colonization resistance in the context of environmental, host, and microbial determinants. Woelfel S, Silva MS, Stecher B.

<https://pubmed.ncbi.nlm.nih.gov/38870899/>

- Additionally, here is a list of additional references that could be added when referring to the effect of antibiotics on human microbiota and susceptibility to pathogens:

<https://pubmed.ncbi.nlm.nih.gov/27707993/>

Short- and long-term effects of oral vancomycin on the human intestinal microbiota - PubMed

Vancomycin-resistant Enterococcus domination of intestinal microbiota is enabled by antibiotic treatment in mice and precedes bloodstream invasion in humans - PubMed

<https://pubmed.ncbi.nlm.nih.gov/21099116/>

Review: Microbiota-mediated colonization resistance against intestinal pathogens - PubMed

<https://pubmed.ncbi.nlm.nih.gov/24096337/>

- In the section on the impact of antibiotics on the gut microbiota in vivo, I recommend that the manuscript: The TaxUMAP atlas: Efficient display of large clinical microbiome data reveals ecological competition in protection against bacteremia.

Schluter J, Djukovic A, Taylor BP, Yan J, Duan C, Hussey GA, Liao C, Sharma S, Fontana E, Amoretti LA, Wright RJ, Dai A, Peled JU, Taur Y, Perales MA, Siranosian BA, Bhatt AS, van den Brink MRM, Pamer EG, Xavier JB. Cell Host Microbe. 2023 Jul 12;31(7):1126-1139.e6. doi: 10.1016/j.chom.2023.05.027. Epub 2023 Jun 16.

is also discussed/ mentioned. In this study the authors observed that, unlike what is often assumed, microbiota communities with high diversity can have carry a broader range of antibiotic resistance genes, than low diversity dysbiotic communities from patients exposed to cancer therapies, because these low diversity dysbiotic communities are often composed of domination of a few species, and domination states can also lead to a reduce diversity in antibiotic resistant genes.

- Line 114, where it is written "bacterial presence", shouldn't it be "bacterial present"

- Line 361, where it is written "host cell physiology", don't you mean on the bacterial physiology?
- Lines 400-403, since this is referring to work that is not published yet, it would be better to elaborate a bit more on what is happening in the processes described here and speculate on the implications.

We thank the reviewers for their positive assessment of our review manuscript and for their constructive comments. We have incorporated their feedback to improve the manuscript. Please find below a detailed response to each point raised.

Referee #1

The review by Müller et al., focuses on implications of antibiotics on the gut microbiome and underlying mechanisms. Overall, I appreciate the breadth and depth of manuscript but would suggest to expand the paragraphs on global differences in antibiotics usage and associations with dysbiosis to better align with the title of the paper. Are there indications that increased antibiotic usage is associated with depletion of key taxa in the population?

If we understood the suggestion from referee 1 properly, they propose to add links between antibiotic usage (quantity and antibiotic classes used) and specific states of dysbiosis. While this would be very interesting, we are not aware of studies that address these questions directly. This is probably because in many countries, antibiotic usage is not regulated and/or faithfully reported. In addition, different states of dysbiosis between different geographic populations might also be caused by other confounding factors (nutrition, environment, etc.). However, there are many studies that sequence the microbiome composition after a given antibiotic treatment, to identify the key taxa that are affected by the treatment. We chose not to list those, as it can be quite long and not relevant for most readers. We now refer the readers to a recent review where they can find this information (line 52-54).

Along the same line it may be prudent to discuss the presence of antibiotic resistance genes in more ancestral populations that may not yet have been exposed to antibiotics.

We now added a few sentences (and corresponding references) on the fact that the presence of antibiotic resistance genes predates antibiotic clinical use (line 290-292).

The paper may also gain broader interest if there is a discussion/outlook on what the options for us a society can do since it is unlikely that antibiotic resistance genes will disappear even if the antibiotic is withdrawn. Beside reducing antibiotic usage there may be alternative approaches such as lantabiotics or phages.

We added a few sentences on the potential use of phages or lantibiotics to decrease dysbiosis (line 192-197). Because this field only starts to be explored, we chose not to extend too much on this part.

Referee #2

In the manuscript "Antibiotic-induced gut microbiome dysbiosis: Risks and strategies for mitigation," Muller and colleagues wrote a review on the recent literature on how antibiotics affect the microbiota and include a critical review on novel recent approaches focusing on understanding the mechanisms behind the effects of antibiotics on the microbiota community and the emergence of antibiotic resistance. They review the work showing how studies comparing laboratory cultures, monocultures, and communities can help identify novel mechanisms and predict the impact of antibiotics on gut communities in animal studies, and discuss the advantages and limitations of the different approaches to studying the impact of antibiotics on the gut microbiota. They also review the literature on how the gut microbiota can serve as a reservoir for antimicrobial resistance and discuss potential strategies to leverage the microbiota to counter-select for the emergence of antimicrobial resistance.

The review is well-written and timely.

We thank the reviewer for these positive comments.

I have only a couple of minor comments/suggestions that can help improve the manuscript:

- Rephrase the sentence on lines 34-35. Here, you should not be saying that bacteria can have different susceptibility profiles. You should change this sentence to make it explicit that gut bacteria can have susceptible profiles that are difficult to predict, given the profiles of typical model bacterial laboratory isolates.

We have rephrased lines 34-35 to explicitly state that gut bacteria display susceptibility profiles that are difficult to predict because they are rarely in line with those of commonly used model bacteria.

- When referring to the role of the microbiota in direct competition to pathogen colonization (colonization resistance), lines 70-72, references to colonization resistance are missing. You can refer to the recent review by Stecher, or add some of the original work reference within that reference. Intestinal colonization resistance in the context of

environmental, host, and microbial determinants. Woelfel S, Silva MS, Stecher B. <https://pubmed.ncbi.nlm.nih.gov/38870899/>

We thank the reviewer for pointing out this omission. We have added the reference by Woelfel, Silva and Stecher to the section on colonization resistance (line 85).

- Additionally, here is a list of additional references that could be added when referring to the effect of antibiotics on human microbiota and susceptibility to pathogens:

<https://pubmed.ncbi.nlm.nih.gov/27707993/> Short- and long-term effects of oral vancomycin on the human intestinal microbiota – PubMed

Vancomycin-resistant Enterococcus domination of intestinal microbiota is enabled by antibiotic treatment in mice and precedes bloodstream invasion in humans - PubMed

<https://pubmed.ncbi.nlm.nih.gov/21099116/>

Review: Microbiota-mediated colonization resistance against intestinal pathogens - PubMed <https://pubmed.ncbi.nlm.nih.gov/24096337/>

We appreciate the suggestions. We have now incorporated these references into the text to broaden our discussion on the effects of antibiotics on human microbiota and susceptibility to pathogens.

- In the section on the impact of antibiotics on the gut microbiota in vivo, I recommend that the manuscript: The TaxUMAP atlas: Efficient display of large clinical microbiome data reveals ecological competition in protection against bacteremia.

Schluter J, Djukovic A, Taylor BP, Yan J, Duan C, Hussey GA, Liao C, Sharma S, Fontana E, Amoretti LA, Wright RJ, Dai A, Peled JU, Taur Y, Perales MA, Siranosian BA, Bhatt AS, van den Brink MRM, Pamer EG, Xavier JB. Cell Host Microbe. 2023 Jul 12;31(7):1126-1139.e6. doi: 10.1016/j.chom.2023.05.027. Epub 2023 Jun 16.

is also discussed/ mentioned. In this study the authors observed that, unlike what is often assumed, microbiota communities with high diversity can have carry a broader range of antibiotic resistance genes, than low diversity dysbiotic communities from patients exposed to cancer therapies, because these low diversity dysbiotic communities are often composed of domination of a few species, and domination states can also lead to a reduce diversity in antibiotic resistant genes.

We thank the reviewer for highlighting this relevant study. We have included a discussion of the TaxUMAP atlas in the section “Prevalence of antibiotic resistance in the human gut microbiome” as we felt it fit best there when discussing the effects of antibiotics on resistance

genes in the context of the loss of diversity in the gut bacterial communities.

- Line 114, where it is written "bacterial presence", shouldn't it be "bacterial present"

We appreciate the reviewer's close reading. However, after reviewing the sentence structure and consulting with the Editor (see the Editor's comment above), we believe "bacterial presence" is correct in this specific context. We have therefore retained the original phrasing (now line 140).

- Line 361, where it is written "host cell physiology", don't you mean on the bacterial physiology?

Thank you for catching this. We intended to refer to bacterial physiology in this context and have corrected the text accordingly (now line 417).

- Lines 400-403, since this is referring to work that is not published yet, it would be better to elaborate a bit more on what is happening in the processes described here and speculate on the implications.

We have expanded this section to provide more detail on the fitness effects associated with resistance determinants that are only exerted in specific communities. As the study is now available on BioRxiv, we have also included the corresponding reference.

Dear Dr. Goemans,

I sincerely apologise for the slow processing of your revised review article due to the high number of primary article submissions that we have to treat with priority. I have now gone through the provided figures, and I think they are suitable for publication in their current form. I am now pleased to inform you that your review has been accepted for publication in The EMBO Journal.

Before we forward your manuscript to our typesetters, I would like to propose some minor edits for style and clarity throughout the manuscript. Please take a look in the attached file and accept or modify as needed.

Your manuscript will be processed for publication by EMBO Press. It will be copy edited and you will receive page proofs prior to publication.

If you have any questions, please do not hesitate to contact the Editorial Office or me directly. Thank you once more for this timely contribution to The EMBO Journal, and I look forward to receiving your input on the final textual edits.

With best wishes,

Ieva

Please note that it is The EMBO Journal policy for the transcript of the editorial process (containing referee reports and your response letters) to be published as an online supplement to each paper. If you should prefer removal of any referee-only figures included in the point-by-point response(s), e.g. because they may still be used for future publication or because they have been reproduced from published work by others, please do let us know immediately via response email.

More information is available here: <https://link.springer.com/partners/embo-press/editorial-policies#Peer%20review>